# Studying miRNA–mRNA Interactions: An Optimized CLIP-Protocol for Endogenous Ago2-Protein

**DOI:** 10.3390/mps5060096

**Published:** 2022-11-30

**Authors:** Sophie Stebel, Janina Breuer, Oliver Rossbach

**Affiliations:** Institute of Biochemistry, Faculty of Biology and Chemistry, University of Giessen, Heinrich-Buff-Ring 17, D-35392 Giessen, Germany

**Keywords:** CLIP, iCLIP, Ago2-CLIP, argonaute-2, microRNA, protein–RNA interaction, RNA-binding protein, UV crosslinking, RNA-seq

## Abstract

Transcriptome-wide analysis of RNA-binding partners is commonly achieved using UV crosslinking and immunoprecipitation (CLIP). Individual-nucleotide-resolution CLIP (iCLIP)enables identification of the specific position of the protein–RNA interaction. In addition to RNA-binding proteins (RBPs), microRNA (miRNA)–mRNA interactions also play a crucial role in the regulation of gene expression. Argonaute-2 (Ago2) mediates miRNA binding to a multitude of mRNA target sites, enabling the identification of miRNA–mRNA interactions by employing modified Ago2-CLIP protocols. Here, we describe an Ago2-specific CLIP protocol optimized for the use of small quantities of cell material, targeting endogenous Ago2 while avoiding possible methodological biases such as metabolic labeling or Ago2 overexpression and applying the latest advances in CLIP library preparation, the iCLIP2 protocol. In particular, we focus on the optimization of lysis conditions and improved radioactive labeling of the 5′ end of the miRNA.

## 1. Introduction

UV crosslinking and immunoprecipitation (CLIP) is a broadly used method to analyze interactions of RNA-binding proteins on a transcriptome-wide scale [1,2]. Via UV-A or UV-C photocrosslinking—the central feature of CLIP—covalent bonds between RNA and their protein interaction partners are formed; through protein-specific precipitation, these interactions can be analyzed on a transcriptome-wide level [3]. Several improvements and variations of the original CLIP protocol have been described since the first CLIP publication [4]; for example, [5] provides a comprehensive review and comparison of the most important CLIP variations. Broadly used adaptions allowing for optimized workflow for various purposes include enhanced CLIP (eCLIP) [6]; crosslinking and sequencing of hybrids (CLASH), which is highly valuable for host RNA–viral RNA interaction analyses [7]; high-throughput sequencing CLIP (HITS-CLIP) [8]; and covalent ligation of endogenous Argonaute-bound RNAs (CLEAR-CLIP) [9]. Additionally, individual-nucleotide-resolution CLIP (iCLIP) further allows for accurate binding site mapping by the terminating reverse transcription at the site of crosslinking [1,10]. Photoactivatable ribonucleoside-enhanced CLIP (PAR-CLIP) involves the incorporation of photoreactive ribonucleoside analogs such as 4-thiouridine (4-SU) in order to increase the crosslinking efficiency of proteins to mRNA targets [11,12,13]. Several publications have shown alternatives to radioactive labeling of crosslinked and immunoprecipitated RNAs, such as PAR-CLIP with direct ligation of fluorescently labeled adapters to crosslinked RNA [14].

Argonaute 2 (Ago2), a component of the RNA-induced silencing complex (RISC), incorporates a multitude microRNAs (miRNA) as guides, each targeting a diverse set of messenger RNA (mRNA). By partially complementarily binding to mRNA, these miRNAs downregulate translation and destabilize the mRNA, significantly impacting post-transcriptional regulation. In the case of completely complementary binding sites, mRNA cleavage and subsequent degradation are initiated, a mechanism often exploited when using small interfering RNAs (siRNA) [15]. In contrast to these inhibiting regulatory functions, miRNA have also been reported to bind to viral RNAs, to enhance translation and replication, and to stabilize the viral RNA genome in hepatitis C virus [16]. Thus, Ago2 is a valuable mediator for the analysis of miRNA–mRNA interactions. The first Ago2-CLIP was conducted with the PAR-CLIP method [17]. Subsequently, Ago2 has been employed, for example, in CLEAR-CLIP [9] and CLASH [7], protocols that allow for interaction analyses of miRNA with cellular mRNAs or viral RNAs [18].

Drawbacks of current Ago2-CLIP protocols range from the large quantities of required cellular starting material to the common use of overexpressed exogenous Ago2 or epitope-tagged endogenous Ago2 in stable cell lines [11,14,17,19,20]. Moore et al. and several other groups have worked with endogenous Ago2 in their studies, mostly using brain tissue of different types of knockout mice or Halo-tagged Ago2 knock-in mice [8,9,20]. As brain tissue consists of several cell types, this may positively affect CLIP efficiency. Few CLIP experiments involving endogenous Ago2 have been described for human cell lines to date. These studies are limited to PAR-CLIP and cells with knockdown of RISC-related proteins potentially influencing Ago2 levels [2,11,18,21,22,23]. Many of these experiments have been further described to require two to five 15 cm dishes per condition and replicate [24]. Kishore et al. additionally described an approach employing an additional micrococcal nuclease (MNase) treatment step to break down large RNA–protein complexes [2,17]. PAR-CLIP further introduces another potential experimental bias by adding 4-SU. This treatment allows for reduced crosslinking energy (365 nm UV-A light instead of 254 nm UV-C) and shorter exposure times, which enables recovery of a higher proportion of miRNA seed-complementary sites [2,25]. Toxicity due to a photoreactive nucleoside has not been observed in PAR-CLIP experiments [12]. However, changes in rRNA levels and reduced cell proliferation upon extended incubation have been reported, which may introduce biases in subsequent analyses [26]. 

Whereas the precipitation of RNA crosslinked to endogenous Ago2 is more challenging, it eliminates biases provoked by Ago2 overexpression or 4-SU addition, thus providing a more authentic snapshot of cellular miRNA–mRNA interactions [27]. This authenticity is of particular relevance for subsequent studies, for example, with respect to the interaction of miRNA with viral transcriptomes [16,18,28,29].

Based on the iCLIP2 protocol described by Buchbender et al. (2020) [1], we focused on overcoming the previously described biases and limitations by reducing the required cellular material and employing immunoprecipitation of untagged, endogenous Ago2 without addition of photoreactive nucleotides. The most important adjustments of this protocol include (i) the reduction in cellular starting materials, (ii) modification of lysis conditions, (iii) improved radioactive labeling of the miRNA 5′ end, and (iv) immunoprecipitation in a cell-line-dependent manner. For new model systems, CLIP-related protocols have to be adjusted and optimized for the respective cell line or tissue. Here, we describe an optimized workflow of the iCLIP2-protocol and highlight the strategically most appropriate adaptations to this methodology.

## 2. Experimental Design

The following instructions are a combination of an analytical approach and a preparative approach to an optimized iCLIP2 protocol employing Ago2 (Figure 1). The analytical approach allows first insights into RNA-binding abilities of a protein or—in the case of Ago2, where RNA-binding ability is established—serves as a time-efficient approach to optimizing CLIP steps. The protocol can be shortened significantly using the analytical approach, as linker-ligations, RNA isolation, and sequencing are omitted. After establishing the optimized protocol, the preparative approach can be employed for RNA isolation and sequencing. 

We found that the Ago2-iCLIP protocol, in addition to relying on the quality of the Ago2 antibody of choice, is also heavily dependent on the cell lines used. We used the Calu3 and an A549 cell lines (both human lung adenocarcinoma-derived). We suggest beginning with the protocol as described below, and if necessary, start by adjusting the lysis duration, salt concentrations of the stringent buffer, and RNase concentrations.

### 2.1. Materials

#### 2.1.1. Reagents

##### General Reagents

HALT protease inhibitors (Thermo Fisher Scientific, Waltham, MA, USA; Cat. no.: 78430);RNaseOUT Recombinant Ribonuclease Inhibitor (Thermo Fisher Scientific, Waltham, MA, USA; Cat. no.: 10777019).

##### Cell Culture, Lysis, and Immunoprecipitation

TURBO DNase (Thermo Fisher Scientific, Waltham, MA, USA; Cat. no.: AM2239);Ambion RNase I (Thermo Fisher Scientific, Waltham, MA, USA; Cat. no.: AM2294);α-Ago2 antibody, clone 11A9, rat monoclonal antibody (Merck KGaA, Darmstadt, Germany; Cat. no.: MABE253);Antibody for negative control, e. g., α-IgG, unconjugated, or α-FLAG;Dynabeads Protein A (Thermo Fisher Scientific, Waltham, MA, USA; Cat. no.: 10002D);Protein G (Thermo Fisher Scientific, Waltham, MA, USA; Cat. no.: 10004D);Sodium chloride (Carl Roth GmbH + Co. KG, Karlsruhe, Germany; Cat. no.: 3957.4).

##### Dephosphorylation and Labeling

Recombinant Shrimp Alkaline Phosphatase (rSAP) (Thermo Fisher Scientific, Waltham, MA, USA; Cat. no.: 783901000UN);T4 polynucleotide kinase (PNK) (New England Biolabs, Ipswich, MA, USA; Cat. no.: M0201L);T4 RNA-ligase (New England Biolabs, Ipswich, MA, USA; Cat. no.: M204L);L3-App 3’ linker (Sequence: /rApp/AGATCGGAAGAGCGGTTCAG/ddC/, DNA adapter is preadenylated at the 5’ end (5′ rApp) to enhance ligation efficiency to crosslinked RNA [1]);Poly (ethylene glycol) (PEG400) (Sigma Aldrich Chemie GmbH, Steinheim, Germany; Cat. no.: 202398);ATP (New England Biolabs, Ipswich, MA, USA; Cat. no.: N0437A);[gamma-32P]-ATP (radioactive concentration: 370 MBq (10 mCi)/mL, specific activity: 29.6 TBq (800 Ci)/mmol) (Hartmann Analytic GmbH, Braunschweig, Germany; Cat. no.: SCP-801).

##### Protein Gel and Transfer

Protein marker VI (pre-stained), peqGOLD (VWR Life Science, Leuven, Belgium; Cat. no.: 27-2310);4× NuPAGE LDS-loading buffer (Thermo Fisher Scientific, Waltham, MA, USA; Cat. no.: NP0007);10× NuPAGE sample-reducing agent (Thermo Fisher Scientific, Waltham, MA, USA; Cat. no.: NP0004).

##### Protein Digestion and RNA Isolation

Proteinase K (20 mg/mL) (Carl Roth GmbH + Co. KG, Karlsruhe, Germany; Cat. no.: 7528.3);RNA-grade phenol/chloroform/isoamyl alcohol (25:24:1) (Carl Roth GmbH + Co. KG, Karlsruhe, Germany; Cat. no.: A156.1);Sodium acetate (Carl Roth GmbH + Co. KG, Karlsruhe, Germany; Cat. no.: 6773.2);GlycoBlue Coprecipitant (Thermo Fisher Scientific, Waltham, MA, USA; Cat. no.: AM9515);2-Propanol (Carl Roth GmbH + Co. KG, Karlsruhe, Germany; Cat. no.: 6752.4);Ethanol (Carl Roth GmbH + Co. KG, Karlsruhe, Germany; Cat. no.: 9065.4).

##### Reverse Transcription

dNTP Mix (VWR Life Science, Leuven, Belgium; Cat. no.: 733-1363);RT1 oligo (Sequence: GGATCCTGAACCGCT, [1]);Superscript III RT (Thermo Fisher Scientific, Waltham, MA, USA; Cat. no.: 56575);0.1 M DTT (Thermo Fisher Scientific, Waltham, MA, USA; Cat. no.: Y00147);Sodium hydroxide (Carl Roth GmbH + Co. KG, Karlsruhe, Germany; Cat. no.: 6771.3);HEPES (Carl Roth GmbH + Co. KG, Karlsruhe, Germany; Cat. no.: 9105.3).

##### Adapter Ligation and PCR

Dynabeads MyONE SILANE (Thermo Fisher Scientific, Waltham, MA, USA; Cat. no.: 37002D);Second 3′ cDNA linker L#clip2.0 (example barcode: /5Phos/NNNNXXXXXXNNNNNAGATCGGAAGAGCGTCGTG/3ddC/with N = random nucleotides and X = experimental barcodes [1]);DMSO (New England Biolabs, Ipswich, MA, USA; Cat. no.: B0515A);High-concentration RNA ligase (New England Biolabs, Ipswich, MA, USA; Cat. no.: M0437);50% PEG8000 (New England Biolabs, Ipswich, MA, USA; Cat. no.: B1004S);2× Phusion HF PCR MasterMix (New England Biolabs, Ipswich, MA, USA; Cat. no.: M0531S);P5 Solexa (sequence: AATGATACGGCGACCACCGAGATCTACACTCTTTCCCT ACACGACGCTCTTCCGATCT [1]);P3 Solexa (sequence: CAAGCAGAAGACGGCATACGAGATCGGTCTCGGCATT CCTGCTGAACCGCTCTTCCGATCT [1]);P5 Solexa_s (sequence: ACACGACGCTCTTCCGATCT [1]);P3 Solexa_s (sequence: CTGAACCGCTCTTCCGATCT [1]);ProNex Size-Selection Purification System (Promega Corporation, Madison, WI, USA; NG2001);Rotiphorese Gel 40 (19:1) (Carl Roth GmbH + Co. KG, Karlsruhe, Germany; Cat. no.: 3030.1);DNA loading Gel Loading Dye Purple (6×) (New England Biolabs, Ipswich, MA, USA; Cat. no.: B7024S) or any other 6× TBE DNA loading buffer;GeneRuler Low-Range DNA Ladder (Thermo Fisher Scientific, Waltham, MA, USA; Cat. no.: SM1191);Ethidium bromide solution 1% (Carl Roth GmbH + Co. KG, Karlsruhe, Germany; Cat. no.: 2218.2).

#### 2.1.2. Buffers 



 Store all buffers at 4 °C and ensure all materials are RNase-free prior to use.

##### General

RNase-free water;Low-salt wash buffer (**TBS-T**) (50 mM Tris-HCl pH 7.4, 150mM NaCl, 0.05% Tween-20);PNK wash buffer (70 mM Tris/HCl pH 7.5, 10 mM MgCl_2_, 0.05% NP-40).

##### Cell Culture, Lysis, and Immunoprecipitation 

Lysis buffer for cytoplasmic extract (10 mM Tris/HCl pH 7.4, 150 mM NaCl, 2 mM EDTA, 1% NP-40 substitute, freshly added: protease inhibitor cocktail);

Or RIPA buffer for nuclear extract (50 mM Tris/HCl pH 7.4, 150 mM NaCl, 5 mM EDTA, 1% NP-40 substitute, 0.1% SDS, freshly added: protease inhibitor cocktail).



 Ensure the addition of protease inhibitor cocktail immediately before use.

RQ1 buffer (40 mM Tris/HCl pH 8, 10 mM MgSO_4_, 1 mM CaCl_2_);High-salt wash buffer (**TBS600/800/1000-T**) (50 mM Tris-HCl pH 7.4, 600/800/1000 mM NaCl, 0.05% Tween-20).

##### Dephosphorylation and Labeling

Linker wash buffer (**TBS400-T**) (50 mM Tris-HCl pH 7.4, 400 mM NaCl, 0.05% Tween-20);PNK dephosphorylation reaction buffer, pH 6.5 (350 mM Tris/HCl pH 6.5, 50 mM MgCl_2_, 5 mM DTT).

##### Protein Gel and Transfer

20× NuPAGE MOPS SDS running buffer (50 mM MOPS, 50 mM Tris base, 0.1% SDS, 1 mM EDTA, pH7.7 (do not adjust pH));20× NuPAGE transfer buffer (25 mM Bicine, 25 mM Bis-Tris (free base), 1 mM EDTA, pH 7.2 (do not adjust pH)).



**Note:** Add 20% Methanol in 1× NuPAGE transfer buffer.

##### Protein Digestion and RNA Isolation

CLIP-PK buffer (100 mM Tris/HCl pH7.4, 50 mM NaCl, 10 mM EDTA, 1% SDS);PK-Urea buffer (100 mM Tris/HCl pH7.4, 50 mM NaCl, 10 mM EDTA, 7 M Urea).

##### Adapter Ligation and PCR

Buffer RLT lysis buffer (QIAGEN GmbH, Hilden, Germany; Cat. no.: 79216).

##### Western Blot

Powdered milk (Carl Roth GmbH + Co. KG, Karlsruhe, Germany; Cat. no.: T145.3);Normal rat IgG-HRP antibody (Santa Cruz Biotechnology Inc., Heidelberg, Germany; sc-2750);Lumi-Light Plus Western Blotting Substrate (Roche Diagnostics GmbH, Mannheim, Germany; Cat. no.: 12015196001).

### 2.2. Equipment 

#### 2.2.1. General Devices

Cooling centrifuge;Scales;Thermomixer;DynaMag-2 Magnet (Thermo Fisher Scientific, Waltham, MA, USA; Cat. no.: 12321D);Thermocycler;Typhoon FLA 9500 biomolecular imager (GE Healthcare GmbH, Munich, Germany; Cat. no.: FLA 9500);Curix 60 Photo Developer (Agfa-Gevaert Group, Mortsel, Belgium; Cat. No.: Curix 60).

#### 2.2.2. Cell Culture, Lysis, and Immunoprecipitation 

Fifteen cm cell culture dishes (Sarstedt AG & Co. KG, Nümbrecht, Germany, Cat. no.: 83.3903.300);SafeSeal reaction tube, 1.5 mm, PP, PCR performance-tested, Low protein-binding (Sarstedt, Nümbrecht, Germany; Cat no.: 72.706.600);Bio-Link BLX-254 UV-Crosslinker (Vilber Lourmat, Eberhardzell, Germany, Cat. no.: BLX-254);Scraper;Branson Sonifier 250 analog ultrasonic cell disruptor with a 1/8” catenoidal tip (BRANSON Ultrasonic Corporation, Danbury, CT, USA; Cat. no.: Branson Sonifier 250; tip Cat. no.: 609-002-021).

#### 2.2.3. Protein Gel and Transfer

XCell SureLock Mini-Cell and XCell II Blot Module (Thermo Fisher Scientific, Waltham, MA, USA; Cat. no.: EI0002);Power supply;NuPAGE Novex 4–12% Bis-Tris Protein gels, 15-well, 1.0 mm (Thermo Fisher Scientific, Waltham, MA, USA; Cat. no.: NP0323BOX);NuPAGE Novex 4–12% Bis-Tris Protein gels, 15-well, 1.5 mm (Thermo Fisher Scientific, Waltham, MA, USA; Cat. no.: NP0336BOX);Amersham Protran 0.45 µm NC nitrocellulose blotting membrane (GE Healthcare GmbH, Munich, Germany; Cat. no.: 10600062);Whatman paper (Bio-Rad Laboratories, Inc., Feldkirchen, Germany; Cat. no.: 1703960);Amersham Hyperfilm MP high-performance autoradiography film (GE Healthcare GmbH, Munich, Germany; Cat. no.: 28906843);

Or

Fuji BAS-SR2040 imaging plate (FUJIFILM Europe GmbH, Ratingen, Germany; Cat. no.: BAS-SR2040). 

#### 2.2.4. Protein Digestion and RNA Isolation

Scalpel;Two mL tubes (Eppendorf AG, Hamburg, Germany; Cat. no.: 0030123344);Phase Lock Gel Heavy 2 mL tubes (QuantaBio; Beverly, MA, USA; Cat. no.: 2302830).

#### 2.2.5. Reverse Transcription

PCR tubes (Nerbe plus, Winsen/Luhe, Germany; Cat. no.: 04-032-0500)

#### 2.2.6. Western Blot

Amersham Hyperfilm ECL high-performance chemiluminescence film (GE Healthcare GmbH, Munich, Germany; Cat. no.: 28906837).

## 3. Procedure

The procedure is described for an analytical, as well as a preparative approach, as visualized in the experimental flow chart presented in Figure 1. 



 For a general setup of the experiment, make sure to plan several controls appropriately, as shown in Figure 2. Proper labeling minimizes mix-up of samples and simplifies tracking of the samples and treatments.

First steps in setting up a CLIP experiment should include preliminary experiments, which are further described in the following protocol. The most prudent adjustments are shown in Figure 3 below.

### 3.1. Bead Preparation for Immunoprecipitation; Time for Completion: 2.5 h

Beginning with Section 3.1. Bead preparation reduces the overall incubation time. During bead antibody incubation, lysate preparation (Section 3.2.) can be conducted, followed by DNase and partial RNase digestion (Section 3.3). For α-Ago2 clone 11A9, use Protein G Dynabeads. If another antibody is used, choose Protein A or G Dynabeads in an antibody-specific manner. Use the same Dynabeads for negative controls (e.g., α-FLAG or α-IgG antibody). 


**Note:** For an analytical approach, up to 50% of the volumes suffice, and recommended volumes differ depending on the approach.

Wash the bead suspension with 2 × 1 mL cold TBS-T.Analytical approach: Use 20 µL bead suspension per reaction.Preparative approach: Use 50 µL bead suspension per reaction.Add monoclonal α-Ago2 antibody to beads. 


**Note:** The smaller volumes listed below have been shown to suffice, but increasing the α-Ago2 antibody amounts may increase the pulldown efficiency for some cell lines.Analytical approach: For every 20 µL of beads, add 0.5–1 µg α-Ago2 antibody or control antibody and 50 µL TBS-T.Preparative approach: For every 50 µL of beads, add 1–2 µg of α-Ago2 antibody or control antibody and 200 µL TBS-T.OPTIONAL ADJUSTMENT An antibody sandwich method can be used by incubating beads with ~2 µg unconjugated α-IgG (rat) antibody for 1 h, washing 2× with cold TBS-T, and subsequently incubating with ~0.5–1 µg α-Ago2 antibody for 1 h according to an analytical approach (see Section 5.3) [27]. This may increase Ago2-IP efficiency in some systems.Incubate on a wheel for 2 h at 4 °C. Continue with Section 3.2 and Section 3.3.

### 3.2. Lysate Preparation; Time for Completion: Harvesting: 10–15 min Per Plate; Lysis: 1−2 h

#### 3.2.1. Pre-CLIP Cell Treatment

The CLIP protocol was optimized in the Calu3 and A549-ACE2 cell lines (both human lung adenocarcinoma-derived) cultured in DMEM with 10% FCS at 37 °C supplemented with 5% CO_2_ for up to 35 passages.

One or several 10–15 cm dishes of cells can be used for one CLIP experiment. Usually, one 15 cm dish of 70% confluence is sufficient for two to six subsequent samples (depending on the volume used for IP). OPTIONAL STEP Adding 100 µM 4-thiouridine (4-SU) to the cells 14 h before harvest increases the efficiency of Ago2-CLIP, as shown in several previous studies, but may introduce additional biases for subsequent analyses [12,26]. The following experimental setup is optimized for lysates without 4-SU treatment. 


**Note**: This protocol is designed for adherent cell lines. Thus, a crosslinking procedure needs to be adapted for non-adherent cells.

#### 3.2.2. Crosslinking of Cells

Cells are crosslinked to covalently bind RNA to proteins in direct contact, providing a snapshot of RNA–protein interactions within living cells [8,27]. We suggest harvesting cells of 70 to 80% confluence. Excessively dense growth has been observed to decrease crosslinking efficiency.
1.Remove the medium by aspiration;2.Wash cells 2 times with ice-cold 1× PBS, and aspirate between washes;3.Remove all liquid before crosslinking;4.Place culture dish in a tray with ice water for cooling and remove lid to crosslink cells under UV light according to Table 1 below.

**Note:** Cells that grow in dense clusters may require increased UV irradiation (we recommend no more than 450 mJ/cm^2^) (Figure 3).5.Add 1 mL 1× PBS to cells, scrape to harvest, and collect in tube.

 CRITICAL STEP: Keep cells on ice unless stated otherwise;6.Centrifuge for 5 min at 2000 rpm at 4 °C;7.Remove all liquid.8.
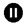
 PAUSE STEP: Continue directly with Section 3.2.3; otherwise, snap freeze in liquid nitrogen and store at −80 °C until further use.

#### 3.2.3. Cell Lysis

To disrupt as many unwanted protein–protein and protein–RNA interactions as possible, a stringent buffer containing ionic detergents is commonly used for cell lysis [3]. Here, we employed lysis buffer (non-ionic detergent) and RIPA buffer (ionic and non-ionic detergents). In accordance with Figure 3, preliminary testing should be conducted for the cell line and protein of interest.

Add 4 packed cell volumes (mg of pellet) of lysis buffer.

**Note:** When interested in the cytosolic Ago2 fraction, the use of lysis buffer is generally a favorable approach (Figure 4). RIPA buffer may improve the lysis of some cell lines and should be used for cases in which nuclear extracts are relevant. The required lysis conditions may differ depending on the cell type. Lysis buffer modifications include:The addition of non-ionic SDS detergent (as described for RIPA buffer), which may improve lysis, in particular that of the nuclear fraction). The Addition of 0.5% sodium deoxycholate is a common approach to further dissociate and solubilize membrane-associated proteins [1,30]. For nuclear extracts, the addition of 1% Triton X-100 may be considered, which strongly permeabilizes the cell membrane and thus improves protein release [14];The addition of 0.5–1 mM ß-mercaptoethanol or dithiothreitol (DTT), which may inhibit the formation of large complexes and therefore improve immunoprecipitation [14,31];Increased salt concentrations of up to 300 mM NaCl in the lysis buffer, which may further support lysis efficiency by increasing the osmotic pressure [31];Control for the cell lysis efficiency using 1× PBS instead of Tris-based buffer [9].OPTIONAL STEP Sonicate cells in a Falcon tube 2 × 15 s (duty cycle/pulse 40%, output control/amplitude 5) to disrupt large complexes and increase yield.

**Note:** This is generally used for bacterial and plant cells but may support the lysis of more resilient cells. The lysis efficiency should be tested in advance. For these first experiments, pellets and non-bound fractions should be included (Figure 4). If the Ago2 Western blot signal is missing or low in the supernatant after lysis and most Ago2-protein is in the pellet after centrifugation of cell debris, sonication should be considered.Depending on the sonicator, at least 1 mL of suspension may be required for sonication; thus, the sample size should be adjusted using additional input material or by increasing the lysis buffer volume per weight;If the pellet is resuspended in RIPA buffer, the addition of detergents is suggested only after sonication to reduce foam formation.Incubate on ice for 10 min and vortex vigorously every 3 min.OPTIONAL ADJUSTMENT: Extension of lysis incubation time from 10 min to ≥30 min on ice and vortexing every 10 min may improve protein concentrations for some cell lines (data not shown) [32].Centrifuge for ≥20 min at a minimum of 13,000 rpm at 4 °C.

 Transfer lysate to SafeSeal reaction tubes and continue to work in these low-protein binding tubes.

**Note:** Ensure that the lysate volume (e.g., between low and high RNase) is distributed equally between all samples. The sample volume for IP should be between 100 and 250 µL for the analytical approach and at least 200 µL for the preparative CLIP approach.Take 5–10% input material for Western control and add 4× LDS-loading buffer, 10× reducing agent, and water. Store at −20 °C until further use.

### 3.3. DNase and Partial RNase Digestion; Time for Completion: 20 min

Predetermine the appropriate RNase I concentrations according to an analytical approach. This should be done after adjustment of appropriate salt concentrations as described in Section 3.4.2. (Figure 3). We suggest a test experiment with a dilution series of RNase concentrations as provided in Table 2; expected results are documented in Section 4.

For each IP sample, add 2 volumes of RQ1 buffer to the extracts;Add 1:500 volume TURBO DNase and 1:1000 RNaseOUT to each sample;Prepare RNase I dilutions in RQ1 buffer (Table 2); RNase I is used because it has no sequence specificity, reducing possible biases in library complexity [3,10];If higher concentrations (e.g., 10^−2^) of RNase are required, the more cost-efficient RNase A can be used instead of RNase I.

**Note**: If the mix is prepared with RNase A instead of RNase I, RNaseOUT cannot be added to the mix because it inhibits RNase A.

**Note**: For Ago2-CLIP with α-Ago2 Clone 11A9 in Calu3 and A549-ACE2 cells, we achieved satisfactory results with high (10^−3^) to low (10^−5^) dilutions.Add 1:100 volume of RNase I dilution to each sample and incubate for 3 min at 37 °C, shaking at 800 rpm. Fill the wells of thermomixer with water for more efficient heat conductivity;Immediately put on ice for ≥3 min.

### 3.4. Immunoprecipitation; Time for Completion: 2.5−3.5 h

#### 3.4.1. Continued Bead Preparation

Wash the bead–antibody suspension 2 times with cold TBS-T;Remove all liquid.

#### 3.4.2. Immunoprecipitation

Precipitate Ago2-RNA complexes from the lysate.



**Note:** TBS800-T was found to be most efficient for stringent washing in these experiments, but the salt concentration can be verified by an IP test or by an analytical approach by comparison with, for example, TBS-600/800/1000-T (600/800/1000 mM NaCl) (Figure 3). 

Add 1:50 volume of 5 M NaCl (final concentration of 150 mM) to each lysate;Add extract to prepared beads and rotate at 4 °C for 2 h;OPTIONAL STEP: Transfer the non-bound fraction to a new tube for Western blot controls and store at −20 °C until further use;Wash 4 times at 4 °C using TBS800-T; during the 3rd wash, transfer beads to fresh tubes. This prevents the elution of complexes unspecifically bound to the plastic of the tube when incubating with sample loading buffer. This may reduce the background when no SafeSeal reaction tubes are used;Wash 2 times with cold PNK wash buffer.

### 3.5. On-Bead Phosphatase Treatment (5′ End); Time for Completion: 1 h

Ago2 was previously suggested to fold around the 5′ end of the incorporated miRNA, thus reducing accessibility for the T4 polynucleotide kinase (T4 PNK) and incorporation of [gamma-32P]-ATP [27]. We found that 5′ end dephosphorylation using recombinant shrimp alkaline phosphatase (rSAP) prior to labeling improved radioactive labeling in this approach, in particular for Ago2-miRNA–mRNA complexes (red arrow in Figure 5).

Prepare 1× phosphatase mix (Table 3):

**Table 3 mps-05-00096-t003:** 1× Phosphatase Reaction Mix.

Volume Per Reaction (µL)	Reagent
4.00	10× phosphatase buffer
1.50	rSAP (1 U/µL)
0.50	RNaseOUT (40 U/µL)
34.00	RNase-free water
Total volume: 40.00	

2.Add 40 µL phosphatase mix to beads and incubate at 37 °C for 20 min;3.Wash 2 times with TBS400-T and transfer to fresh tube during 2nd wash;4.Wash 2 times with PNK wash buffer.


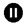
 PAUSE STEP: Analytical approach: Store beads in PNK wash buffer overnight at 4 °C or continue with Section 3.6.

### 3.6. On-Bead Phosphatase Treatment and 3′-RNA Linker Ligation; Time for Completion: 17 h

Linker ligation can be omitted in the analytical approach if samples are not further processed for sequencing but only analyzed on a nitrocellulose membrane. 


**Note:** The additional washing steps during linker ligation further reduce background signal in radiography, thus simplifying Section 3.8.1, i.e., RNA isolation from the CLIP membrane.

#### 3.6.1. Phosphatase Treatment of 3′ Ends

Prepare phosphatase mix to dephosphorylate the 3′ end. The low pH of PNK dephosphorylation buffer enables dephosphorylation by T4 PNK (Table 4);

**Table 4 mps-05-00096-t004:** 1× Dephosphorylation Mix.

Volume Per Reaction (µL)	Reagent
4.00	5× PNK dephosphorylation buffer (pH 6.5)
0.50	T4 PNK enzyme (NEB, with 3′ phosphate activity)
0.50	RNaseOUT (40 U/µL)
15.00	RNase-free water
Total volume: 20.00	

2.Add to beads and incubate in thermomixer at 37 °C and 1000 rpm for 20 min;3.Wash 2 times with TBS400-T; transfer to a fresh tube during the 2nd wash;4.Wash 2 times with PNK wash buffer.

#### 3.6.2. On-Bead 3′ Linker Ligation

Prepare ligase mix (Table 5):

**Table 5 mps-05-00096-t005:** Ligase Mix for On-Bead 3’ Linker Ligation.

Volume Per Reaction (µL)	Reagent
1.00	Preadenylated linker L3-App (20 µM)
2.00	10× T4 RNA-ligase buffer
1.00	T4 RNA-ligase
0.50	RNaseOUT (40 U/µL)
4.00	PEG400
11.50	RNase-free water
Total volume: 20.00	

2.Add ligase mix to beads;


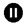
 PAUSE STEP: Incubate at 16 °C and 1000 rpm overnight;

3.Wash twice with cold PNK wash buffer.

### 3.7. Radioactive Labeling, SDS-PAGE, and Nitrocellulose Transfer; Time for Completion: Gel and Transfer: 4−7 h; Exposure: 16−72 h



 From this point forward, conduct experiments in an isotope laboratory with appropriate protective measures.

#### 3.7.1. Radioactive Labeling

Prepare PNK mix (Table 6):



**Note:** Our experience shows that Ago2-CLIP can effectively be conducted with a mix of 1 part [gamma-32P]-ATP and 1 part non-radioactive ATP (same concentrations) per sample. Ratios of [gamma-32P]-ATP and ATP can be adjusted further to reduce or increase the radioactive signal. Whereas a stronger signal reduces exposure times of X-ray films, additional radioactivity will be transferred into the RNA isolation.

Analytical approach: Use 1 µL of ATP mix;Preparative approach: Use 2.5 µL of ATP mix.

2.Add to beads and incubate in thermomixer at 37 °C for 20 min at 1000 rpm;3.Wash once with cold TBS-T;4.Wash once with cold PNK wash buffer;5.Resuspend beads in 1× LDS-loading buffer (4× NuPAGE LDS-loading buffer, 10× NuPAGE Reducing agent, RNase-free water);Analytical approach: Use 20 µL LDS-loading buffer;Preparative approach: Use 30 µL LDS-loading buffer;6.Denature for 10 min at 70 °C at 1000 rpm.

#### 3.7.2. SDS-PAGE



 CRITICAL STEP: It is crucial to use a protein gel system with a neutral pH to prevent RNA degradation by alkaline hydrolysis [1]. The Invitrogen NuPAGE system meets these requirements. Alternatively, other gel systems that retain neutral pH during the electrophoresis run, e.g., as described by Graham et al., can be used [33].

Use a NuPAGE system and 4–12% Bis-Tris gels in 1× NuPAGE MOPS SDS running buffer;Analytical approach: gels with a well size of 1.0 mm are sufficient;Preparative approach: use gels with a well size of 1.5 mm to load more volume and yield a higher RNA amount for sequencing.Load 6 µL prestained protein marker;Load inputs and samples in appropriate volume for well size;Run gel for a minimum of 2 h at 200 V.

**Note:** The relevant area is above the band of the protein of interest (above 100 kDa); thus, the gel run can be extensive. For Ago2-CLIP, a 2 to 3 h gel run is optimal. 

 If analyzing truncated constructs, the duration should be adjusted accordingly.

#### 3.7.3. Transfer

Using the wet transfer chamber approach of the NuPAGE system, gel is transferred onto a nitrocellulose membrane. This is an additional cleanup step, as only RNAs covalently crosslinked to precipitated proteins are transferred. Free RNA does not bind to nitrocellulose with sufficient affinity.

Incubate the nitrocellulose membrane, sponges, and Whatman paper in 1× NuPAGE transfer buffer (supplemented with 20% Methanol) for >20 min before use;Set up the system according to the manufacturer’s protocol;Blot at 30 V for 1 h or at 15 V overnight;Draw marker bands with a radioactive pen onto a piece of plastic bag and place on/next to membrane;Place inside a plastic bag;Expose the membrane;Analytical approach: Expose the membrane to a BAS-SR2040 imaging plate for 16−72 h at room temperature for subsequent visualization in Typhoon FLA 9500;Preparative approach: Expose the membrane one or multiple times to Amersham Hyperfilm MP high-performance autoradiography film for at least 16−72 h at −80 °C and develop in a photo developer. 


**Note:** Exposing two films at once is an effortless method to obtain different exposure levels.

OPTIONAL STEP: Analytical approach: Continue with Western Blot under protective measures in an isotope laboratory.

### 3.8. RNA Isolation and Clean-Up; Time for Completion: 48−72 h

#### 3.8.1. RNA Isolation from CLIP Membrane

Use X-ray film as a reference and cut the region of interest from the CLIP membrane (Figure 6);Add nitrocellulose pieces to 2 mL reaction tubes containing 400 µL CLIP-PK buffer and 20 µL 20 mg/mL proteinase K;Incubate at 37 °C at 1000 rpm for 20 min;Add 400 µL of PK-urea buffer and incubate for an additional 20 min at 55 °C while shaking;Add 800 µL RNA-grade phenol/chloroform/isoamyl alcohol (25:24:1);Incubate for 5 min at 30 °C and 1000 rpm;Separate phases by spinning for 5 min at 13,000 rpm at 4 °C using Phase Lock Gel Heavy 2 mL tubes;Transfer aqueous layer into a new tube without touching the wax;Precipitate by adding 1 µL GlycoBlue and 80 µL 3 M sodium acetate, pH 5.5. Mix and add 0.7 volumes (616 µL) cold isopropanol;
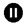
 PAUSE STEP: Store at −80 °C overnight for precipitation;Spin at full speed for ≥30 min at 4 °C;Remove the supernatant, wash pellets with 1 mL 70% ethanol, and spin for 5 min; 
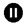
 PAUSE STEP: Store in 70% ethanol at −80 °C until further processing;Dissolve pellets in 12 µL water;OPTIONAL STEP: As a control, perform a Western blot of the remaining nitrocellulose membrane under protective measures in an isotope laboratory.

#### 3.8.2. Reverse Transcription

Whereas miRNAs with a length of 17–24 nt are difficult to reverse transcribe, this can be disregarded here owing to the previous linker ligation, which increases fragment size. Additionally, linker ligation has been reported to mediate the ligation of mRNA interaction partners to Ago2-incorporated miRNA, providing an additional insight with respect to interactions without additional ligation steps [9,18,34]. RNA of the same sample and different RNase concentrations can be pooled and labeled with the same barcode in subsequent steps.

Prepare RNA–primer mix (Table 7):

**Table 7 mps-05-00096-t007:** RNA-Primer Mix for Reverse Transcription.

Volume Per Reaction (µL)	Reagent
12.00	RNA
1.00	dNTP mix (10 mM)
1.00	RT1 oligo (0.5 pmol/µL working solution [stock: 100 µM = 100 pmol/µL])

2.Place in thermocycler: 70 °C 5 min, 25 °C hold until RT mix added. Mix by pipetting; 

 CRITICAL STEP: Never put on ice;3.Prepare RT mix (Table 8):

**Table 8 mps-05-00096-t008:** Reverse Transcription Mix.

Volume Per Reaction (µL)	Reagent
4.00	5× first strand buffer
1.00	0.1 M DTT
0.50	Superscript III RT (200 U/µL)
0.50	RNaseOUT (40 U/µL)
Total volume: 6.00	

4.Add to sample (total volume 20 µL) and place in thermocycler at 25 °C for 5 min, 42 °C for 20 min, and 50 °C for 40 min, and hold at 4 °C;5.Add 1.65 µL 1 M NaOH and incubate at 98 °C for 20 min to degrade the RNA and remove residual radioactive phosphates from the mixture in the subsequent cleanup step;6.Add 20 µL 1 M HEPES-NaOH, pH 7.3;


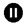
 PAUSE STEP: Store cDNA at −20 °C until further use.

#### 3.8.3. MyONE Cleanup

The following steps of the protocol are adopted from the iCLIP2 protocol developed by Buchbender et al. [1]. All essential steps are described below, but for more detailed explanations of these parts, refer to the original protocol.

Mix magnetic MyONE silane bead solution thoroughly and use 10 µL beads per sample;Wash beads with 500 µL RLT buffer using a magnetic stand and resuspend in 125 µL RLT buffer; add to sample and mix;Add 150 µL 100% ethanol and carefully mix by pipetting; incubate for 5 min at room temperature, mix, and incubate for an additional 5 min;Place beads on a magnetic rack and discard the supernatant;Resuspend beads in 1 mL 80% ethanol and transfer the mix to a new tube;Wash 2 times with 80% ethanol. Incubate 30 sec each time before placing the sample on a magnet;Spin down in a microcentrifuge, place on a magnet, and discard the supernatant;Air dry beads for 5 min at room temperature and resuspend in 5 µL water;Incubate the mix at room temperature for 5 min and 

 proceed without removal of beads;



**Note:** At this point, samples should be free of residual radioactive ATP and RNA fragments. After verifying this, the experiment can be continued outside an of isotope laboratory.

#### 3.8.4. Second Linker Ligation to 3′ End of cDNA

Choose different 3′ cDNA linkers for each experiment and control (Table 9). Barcoding enables pooling of several experiments before sequencing. Using every possible base at every position or using A at every position of the experimental barcode facilitates 2-color Illumina sequencing [1].

Add 2 µL (10 µM) specific second adapter (L#clip2.0, [1]) and 1 µL 100% DMSO to the cDNA–bead solution;Heat the mix for 2 min at 75 °C and immediately put on ice for >1 min;Prepare the ligation mix on ice. Mix by vigorously stirring, pipetting, and flicking to ensure homogeneity. Briefly spin down in a microcentrifuge;Add 12 µL ligation master mix to 8 µL sample-linker mix and mix thoroughly;Add another 1 µL RNA ligase to each sample to a final volume of 21 µL and mix by stirring;
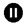
 PAUSE STEP: Agitate overnight at room temperature and 1100 rpm.

#### 3.8.5. MyONE Cleanup 2

Fresh MyONE beads are added to the cDNA–bead mix from the ligation reaction (Section 3.8.4).

Mix magnetic MyONE silane bead solution thoroughly, and use 5 µL beads per sample;Wash beads with 500 µL RLT buffer and resuspend them in 60 µL RLT buffer; add to the sample and mix;Add 60 µL 100% ethanol and carefully mix by pipetting; incubate for 5 min at room temperature, mix, and incubate for another 5 min;Place beads on a magnetic rack and discard the supernatant;Resuspend beads in 1 mL 80% ethanol and transfer the mix to a new tube;Wash 2 times with 80% ethanol; incubate 30 s each time before putting the sample on the magnet;Spin in a microcentrifuge, place on the magnet, and discard supernatant;Air dry beads for 5 min at room temperature and resuspend in 23 µL water;Incubate the mix for 5 min at room temperature;Magnetically attract beads and add eluate to PCR mix of 3.9.

### 3.9. First PCR; Time for Completion: 4−5 h (Additional 2−3 h per Subsequent PCR Test 

#### 3.9.1. cDNA Preamplification

Prepare the PCR mix (Table 10):

**Table 10 mps-05-00096-t010:** PCR Mix for cDNA Preamplification.

Volume Per Reaction (µL)	Reagent
22.50	cDNA
2.50	Primer mix of P5Solexa_s and P3Solexa_s (10 µM each)
25.00	2× Phusion HF PCR MasterMix
Total volume: 50.00	

2.Run PCR (Table 11):

**Table 11 mps-05-00096-t011:** PCR Scheme for Preamplification of cDNA.

Step	Temperature (°C)	Time	Cycles
1	98	30 s	
2	98	10 s	6 cycles
	65	30 s
	72	30 s
3	72	3 min	
4	16	Hold	


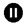
 PAUSE STEP: Store cDNA at −20 °C until further use.

#### 3.9.2. First ProNex Size Selection

This step is conducted to remove primer–dimers.

Equilibrate ProNex chemistry to room temperature for 30 min;Resuspend beads by vortexing vigorously;For a 50 µL sample (PCR product), add 147.5 µL beads (1:2.95 *v*/*v* ratio of sample to beads);Mix by pipetting 10 times up and down;Incubate samples at room temperature for 10 min;Place samples on a magnetic stand for 2 min, and discard the supernatant;Leave beads on the magnetic stand and add 300 µL ProNex wash buffer to the samples; incubate for 30–60 s before removal;

**Note:** Ensure the wash buffer covers all beads on the magnet;

 CRITICAL STEP: Do not remove ProNex beads from magnet or resuspend during the wash; this can cause up to 20% sample loss. For large samples, increase the volume of ProNex wash buffer proportionally to the volume of samples and beads [1];Repeat the wash and allow sample to air dry for 8–10 min (<60 min) until cracking starts;Remove the beads from the magnetic stand and start eluting the samples. Resuspend beads of the samples in 23 µL ProNex elution buffer;Resuspend all samples by pipetting; then, let them incubate for 5 min at room temperature;Return samples to the magnetic rack for 1 min; then, carefully transfer eluted cDNA to a clean tube.


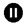
 PAUSE STEP: Freeze at −20 °C until further processing.

#### 3.9.3. Second PCR Amplification—PCR Cycle Optimization—Test PCR 1

The PCR test should be performed with two or three different cycle numbers to determine the optimal cycles for the preparative PCR. For Ago2-CLIP, we recommend 14 to 20 cycles for the first PCR test.

Prepare the PCR mix for two/three cycle numbers (Table 12):

**Table 12 mps-05-00096-t012:** PCR Mix for Second Amplification and Cycle Optimization.

Volume Per Reaction (µL)	Reagent
Mastermix 1	Per sample x cycle number
7.00	Water
1.00	Primer mix of P5/P3 Solexa (10 µM each)
Volume per cycle number: 8.00	
Mastermix 2	Per each reaction
8.00	Mastermix 1
1.00	cDNA
5.00	2× Phusion HF PCR MasterMix
Final volume per reaction: 10.00	

2.Run PCR (Table 13):

**Table 13 mps-05-00096-t013:** PCR Scheme for Second Amplification.

Step	Temperature (°C)	Time	Cycles
1	98	30 s	
2	98	10 s	14, 16, 20 cycles
	65	30 s
	72	30 s
3	72	3 min	
4	16	Hold	



 CRITICAL STEP: At this point, the samples should not be opened in the same working space as before. If possible, switch to a designated post-PCR bench or even room and use different pipettes and equipment. Using the same pipettes as before may contaminate future samples with PCR products of previous experiments, falsifying the sequencing results;

3.Combine 10 µL PCR with 2 µL 6× DNA-loading buffer and load 6 µL;4.Use 0.5–1 µL of GeneRuler Low-Range DNA Ladder diluted in a total volume of 6 µL 1× loading buffer as a size marker, and load alongside samples;5.Run on 7% PAA-TBE gel for 30 min at 200 V and stain for 10 min with ethidium bromide solution.

#### 3.9.4. Second PCR Amplification—PCR Cycle Optimization—Test PCR 2

If cycle numbers need further adjusting, employ the following test PCR.

For each cycle number, mix according to the PCR mix in Section 3.9.3.;Run PCR according to Section 3.9.3. with adjusted cycle numbers and proceed with the TBE gel protocol accordingly.

If necessary, step can be repeated for further optimization.

### 3.10. Preparative PCR; Time for Completion: 4−5 h

#### 3.10.1. PCR

Prepare the PCR mix (Table 14):



**Note:** We advise using only half of your library (10 µL); keep the other half as a backup;

**Table 14 mps-05-00096-t014:** Preparative PCR Mix for Subsequent Sequencing.

Volume Per Reaction (µL)	Reagent
8.00	Water
2.00	Primer mix of P5/P3 Solexa (10 µM each)
20.00	2× Phusion HF PCR MasterMix
30.00	Each
+10.00	cDNA
Total volume: 40.00	

2.Run PCR as previously described in Section 3.9.3 and Section 3.9.4, adjusted to the final cycle number;3.Combine 5 µL PCR with 1 µL 6× DNA-loading buffer;4.Use 0.5–1 µL of GeneRuler Low-Range DNA Ladder diluted in a total volume of 6 µL 1× loading buffer as a size marker, and load alongside samples;5.Run on 7% PAA-TBE gel for 30 min at 200 V and stain for 10 min with ethidium bromide.

#### 3.10.2. Second ProNex Size Selection to Remove Residual Primers

The second cDNA size selection eliminates excess primer that may negatively influence sequencing results. Here, a different sample-to-ProNex bead ratio than in the first size selection is required because primers in preparative PCR are longer in comparison to cDNA preamplification (Table 15).

**Note:** Library preparation can be performed only with samples of interest, excluding negative controls if they are excluded from sequencing.

Equilibrate ProNex beads to room temperature for 30 min and resuspend beads by vortexing vigorously;Based on the optimal sample-to-ProNex (*v*/*v*) ratio of 1:2.4, calculate the required volume of beads for all samples. 


**Note:** Keep in mind that sample volume will be reduced as a result of removal of input;Take reagents to a post-PCR lab (e.g., for 34.5 µL volume per sample):

4.Take input from each sample, and pool in one tube (1% input + water + 1 µL 6× DNA-loading buffer; 6 µL total);5.Add the calculated volume of beads to each sample. Mix by pipetting up and down 10 times;6.Incubate the samples at room temperature for 10 min. Place the samples on a magnetic rack for 2 min;7.Leave beads on magnetic stand and add 300 µL ProNex wash buffer to samples;



**Note:** Ensure that the wash buffer covers all beads on the magnet; incubate for 30–60 s before removal.

8.Wash samples a second time with 300 µL ProNex wash buffer. Remove the wash buffer after 30–60 s;9.Remove the beads from the magnetic rack and start eluting the samples. Be sure to elute beads of the samples in 20 µL ProNex elution buffer;10.Resuspend all samples by pipetting and let them stand for 5 min at room temperature;11.Return the samples to the magnetic stand for 1 min; then, carefully transfer the eluted cDNA to a clean tube;12.Pool half of each sample in one reaction tube;13.Use 1% of library to assess primer removal on a 7% PAA-TBE gel;Combine 1% library with water and 1 µL 6× DNA-loading buffer;14.Analyze on 7% PAA-TBE gel together with previously pooled inputs (step 4) for 25 min to determine primers.

Half of the prepared library can be sent to sequencing facilities for bioanalyzer analyses and sequencing runs.



**Note:** We suggest retaining some of the samples as a backup.

For limited sample and linker numbers, spike in for sequencing is recommended.

## 4. Expected Results

### 4.1. DNase and Partial RNase Digestion

Preliminary testing of RNase I concentrations should be performed to determine the most effective concentrations, as described in Section 3.3. A smear is visible above the protein of interest in the radiography. With lower RNase I concentrations, the smear increases in size, owing to a lower degree of RNA degradation (Figure 5, red box).

### 4.2. Immunoprecipitation

Immunoprecipitation should be tested beforehand to ensure antibody efficiency. Additionally, lysis conditions can be optimized. Figure 4 shows Western blots of preliminary IPs with lysis buffer in comparison with RIPA buffer and including all fractions of immunoprecipitation.

### 4.3. Radiography 

It has been reported that Ago2-miRNA complexes show a radioactive signal at ~110 kDa, whereas Ago2-miRNA–mRNA complexes show a signal at >130 kDa [8,27]. The radiography should show a smear, specifically in the area above the protein of interest. Smear intensity increases with lower RNase I concentrations, as less RNA has been degraded (Figure 5).

In the subsequent preparative approach, the nitrocellulose membrane after transfer is exposed to an X-ray film. The X-ray film serves as a template to cut out the appropriate fragments. A Western blot is performed subsequently to analyze immunoprecipitation efficiency (Figure 6).

After the second PCR amplification test run, cDNA is loaded on 7% PAA-TBE gels. Bands for miRNA should run at around 154 nt, as the total linker length is 134 nt, and miRNA length is ~20 nt [1]. Bands of larger sizes indicate miRNA–mRNA hybrids, as previously visualized by Moore et al. [27].

## 5. Further Explorations and Troubleshooting

As previously mentioned, important adjustments include the salt concentration of stringent washing, cell-line-dependent crosslinking, lysis conditions, and RNase I dilutions (Figure 3). These can be adjusted as described above. Below, we detail antibody restrictions and list further potential modifications.

### 5.1. α-Ago2 Antibody Choice

Choosing an α-Ago2 antibody is not a trivial task, considering the variety of available antibodies. In our experiments, we tested several antibodies with varying results. Precipitation with α-Ago2 antibody clone 9E8.2 (Merck KGaA, Darmstadt, Germany, Cat. no.: 04-642) was functionally limited, and detection was only possible on a PVDF membrane. Because iCLIP2 relies on transfer onto nitrocellulose, this antibody was not further employed. Using the Anti-pan Ago antibody clone 2A8 (Merck KGaA, Darmstadt, Germany, Cat. no.: MABE56) also showed no signals in preliminary IPs. Western blot of α-Ago2 clone 11A9 in 5% milk-TBS-T yielded an adequate protein signal and indicated successful IP; thus, all subsequent experiments were conducted with this antibody.

### 5.2. Depletion of Large Ribonucleoprotein Particles (RNPs)

A step commonly described in immunoprecipitation protocols (after Section 3.4.2 step 1) is the removal of large RNPs by centrifugation for 5 min at 16,400 rpm at 4 °C [1]. This has been shown to substantially reduce the RNA signal in the radiography in our experiments (data not shown). Thus, we advise excluding this step from the immunoprecipitation protocol.

### 5.3. Antibody Sandwich Method

Moore et al. first described a sandwich antibody method to increase immunoprecipitation efficiency in CLIP experiments [27]. Incubation of Dynabeads with α-IgG before incubation with α-Ago2 may reduce the effects of steric hindrance and increase the binding efficiency of α-Ago2 to the beads via the mediating antibody. Additionally, the intermediate offers extra binding sites for α-Ago2, as illustrated in Figure 7.

Whereas Moore et al. applied 50 µg of unconjugated α-IgG (mouse) bridging antibody to 200 µL Dynabeads solution and then added 2 µg Anti-pan Ago antibody clone 2A8 [27], we suggest reducing the ratio of bridging antibody to the α-Ago2 antibody and beginning with an analytical approach. 



**Note**: Choose an appropriate α-IgG antibody for the α-Ago2 antibody depending on the organism of origin (α-rat IgG for α-Ago2 clone 11A9).



**Note**: In our experiments, we did not observe significant improvement using this method. In other cell types or when using different lysis methods, the bridging antibody approach may improve Ago2-IP efficiency [27].

### 5.4. Suggested Explorations

In case further optimization is necessary for the applied cell line, the following options for improvement are applicable.

*Labeling* efficiency may be improved by mixing [gamma-32P]-ATP with ADP instead of ATP [35]. Another option is radioactive 3′ end labeling using [alpha-32P]-ATP and terminal transferases, an approach which has been reported to result in reduced labeling efficiency and thus longer autoradiography exposure times [27,36]. 

The use of radioactivity is not mandatory for CLIP procedures. Protocols have been published using direct ligation of a fluorescently labeled adapter to the 3’ end of crosslinked RNA. Refer to [14] for a detailed protocol on non-radioactive PAR-CLIP for small RNA library preparations.

## Figures and Tables

**Figure 1 mps-05-00096-f001:**
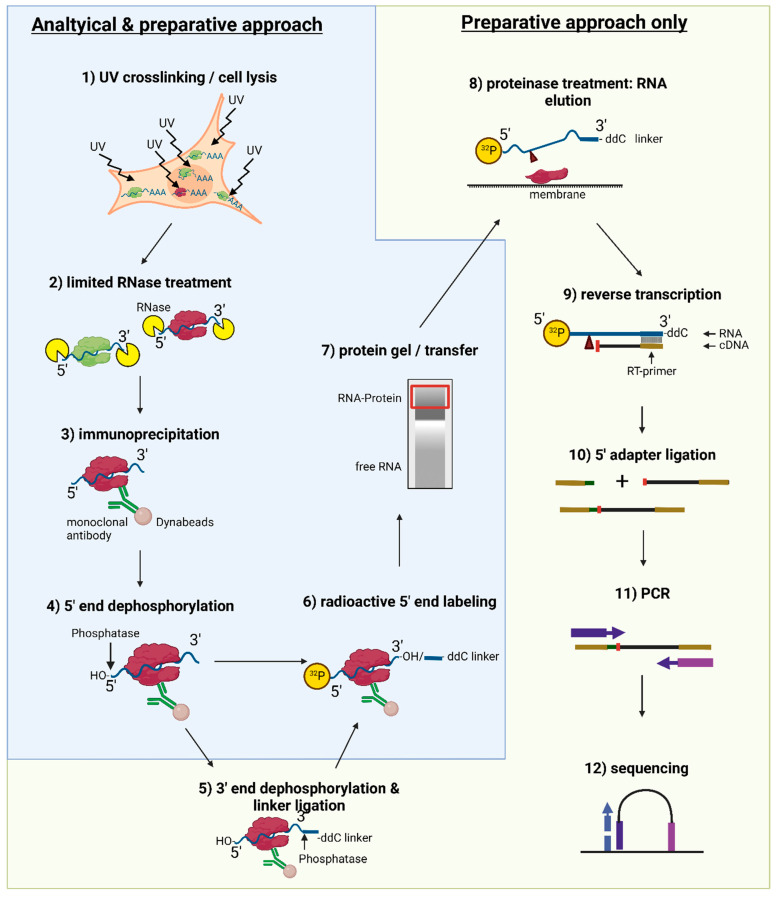
Overview over the iCLIP2 protocol. Experimental steps necessary for both analytical and preparative approaches (UV crosslinking and cell lysis, limited RNase treatment, IP, 5′ end dephosphorylation, radioactive 5′ end labeling, protein gel, and membrane transfer) are indicated by a blue color background and outline. A green background and outline indicate additional steps only necessary for a preparative approach (additional 3′ end dephosphorylation and 3′-linker ligation, proteinase treatment for RNA elution from the membrane, reverse transcription, 5′ adapter ligation, PCR, and sequencing).

**Figure 2 mps-05-00096-f002:**
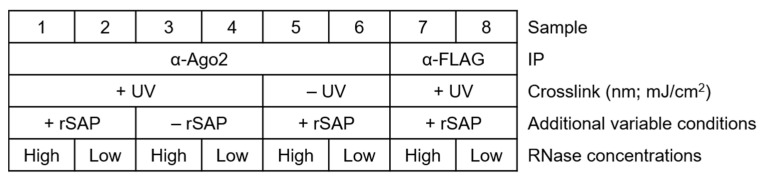
Simplified table showing how to sort samples and their treatments, including important control reactions. Samples with an antibody of interest and their additional variable treatments (crosslinking, rSAP treatments, ligase treatment, and NaCl concentration of stringent buffers), high- and low-RNase I concentrations, and negative controls (α-IgG or α-FLAG antibody for IP).

**Figure 3 mps-05-00096-f003:**
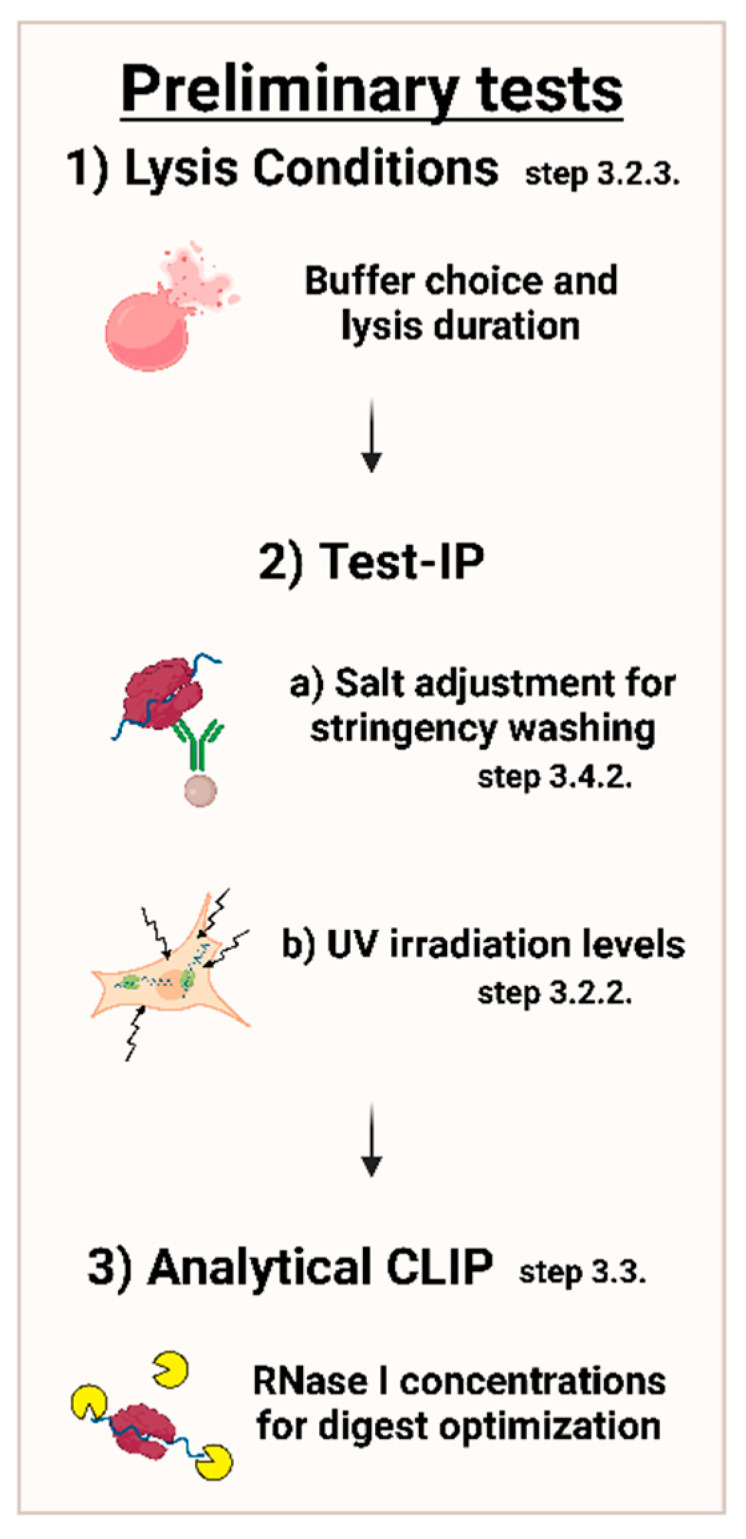
Overview of important conditions to adjust in preliminary tests. Conditions should be adjusted in order of lysis conditions with respect to the buffer system and incubation length, stringency of washing with various salt concentrations, along with UV irradiation, followed by RNase I concentrations.

**Figure 4 mps-05-00096-f004:**
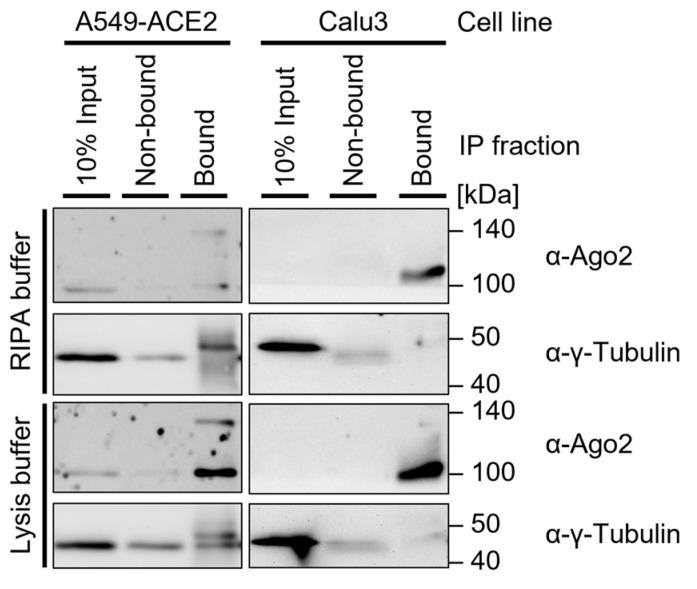
Western blot of immunoprecipitation in two cell lines: A549-ACE2 and Calu3. Both cell lines were UV-crosslinked (300 mJ/cm^2^), lysed in 4× RIPA buffer or lysis buffer, and sonicated (2 × 15 s, duty cycle/pulse 40%, output control/amplitude 5), then incubated with 0.5 µg α-Ago2 Clone 11A9, washed with TBS800-T, and resuspended in LDS-loading dye. Then, 6 µL protein marker VI (pre-stained) peqGOLD and 12 µL samples were loaded on a 15-well 12% SDS-PAGE gel and transferred onto an Amersham Protran 0.45 µm NC nitrocellulose membrane. The membrane was blocked in 5% milk-TBS-T and incubated in 5% milk-TBS-T with an α-Ago2 Clone 11A9 (1:500) and α-γ-Tubulin (1:5000) with the respective secondary antibody, followed by exposure to Amersham Hyperfilm ECL film.

**Figure 5 mps-05-00096-f005:**
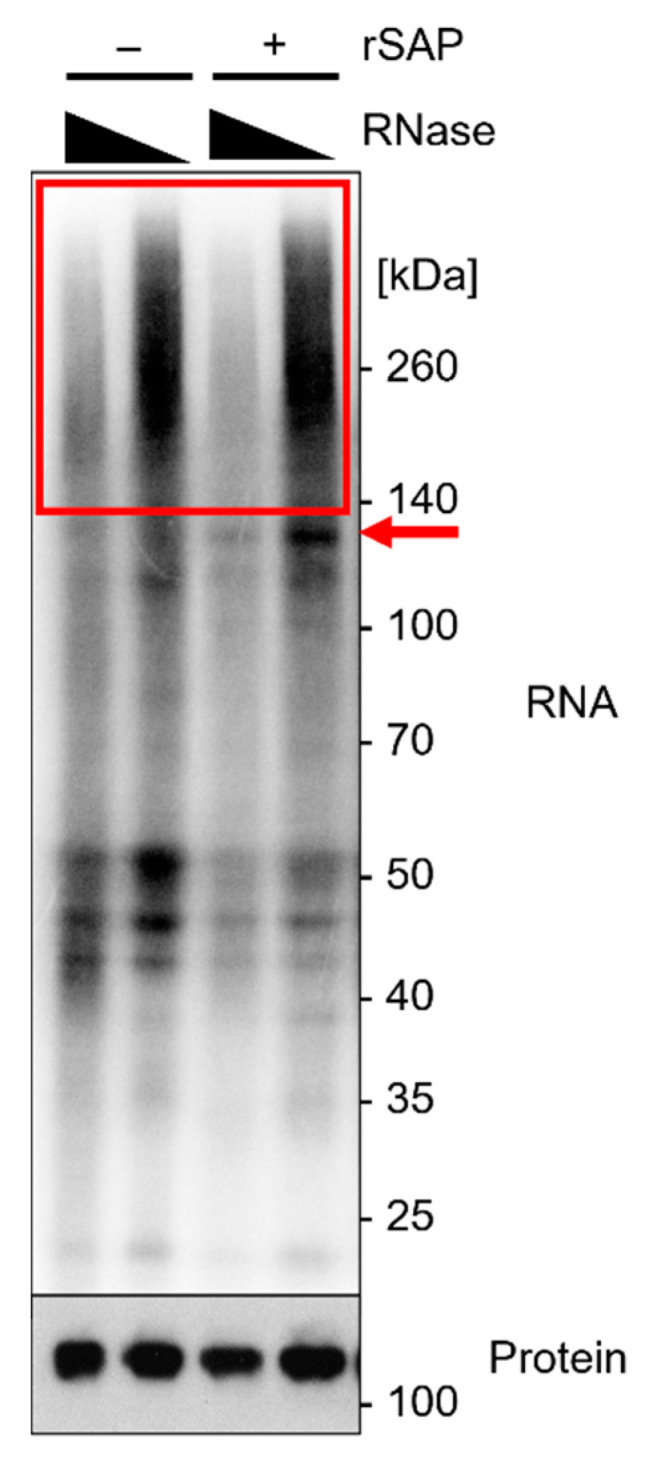
Analytical approach of Ago2-iCLIP2. Radiography of expected results on NuPAGE gel using protein marker VI (pre-stained) peqGOLD. A volume of 10 µL of sample was loaded, the gel was run for 90 min at 200 V, and transferred to an Amersham Protran 0.45 µm NC nitrocellulose membrane for 1 h at 30 V. The sample was subsequently exposed to a Fuji BAS-SR2040 imaging plate overnight. Calu3 cells were crosslinked at 254 nm with 450 mJ/cm^2^ and lysed by sonication in 4× packed cell volume lysis buffer. IP with 0.5 µg α-Ago2 clone 11A9, RNase I digest employing dilutions of 10^−3^ and 10^−5^, and with or without 5′ end dephosphorylation using rSAP before radioactive labeling with ATP mix (1/2 volume [gamma-32P]-ATP). The red box indicates the area of interest with an RNase I concentration-dependent Ago2-bound RNA smear. The red arrow indicates Ago2-miRNA–mRNA complexes, which are more pronounced after 5′ end dephosphorylation. Subsequent Western blot analysis of protein levels of Ago2 on CLIP-membrane, with 1:500 α-Ago2 clone 11A9 as primary antibody in 5% milk-TBS-T and 1:10,000 α-rat IgG-HRP as secondary antibody. Visualization on Amersham Hyperfilm ECL film.

**Figure 6 mps-05-00096-f006:**
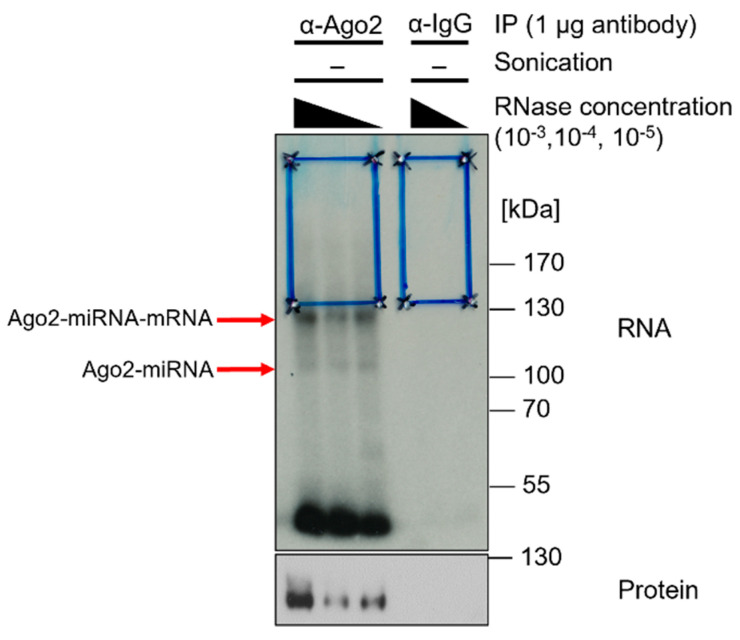
X-ray film of radiography. α-Ago2 clone 11A9 IP and control IP with unconjugatedα-IgG (rat) of Calu3 lysates in lysis buffer crosslinked at 254 nm and 450 mJ/cm^2^, sonicated (2 × 15 s, duty cycle/pulse 40%, output control/amplitude 5), and RNase I-digested at concentrations of 10^−3^, 10^−4^, and 10^−5^. Radioactive labeling with ATP mix (1/5 volume [gamma-32P]-ATP). Samples of 20 µL were loaded with 6 µL protein marker VI (pre-stained) peqGOLD. NuPAGE gel was run for 150 min at 200 V and transferred onto an Amersham Protran 0.45 µm NC nitrocellulose membrane for 1 h at 30 V. Then, the sample was exposed to Amersham Hyperfilm MP high-performance autoradiography film for five days. Blue boxes indicate regions of interest on the membrane, which was cut for further processing. Red arrows indicate Ago2-miRNA–mRNA and Ago2-miRNA complexes. Subsequent Western blot on CLIP-membrane. α-Ago2 clone 11A9 (1:500) in 5% milk-TBS-T, followed by α-rat IgG. Exposure with Lumi-Light Plus on Amersham Hyperfilm ECL film.

**Figure 7 mps-05-00096-f007:**
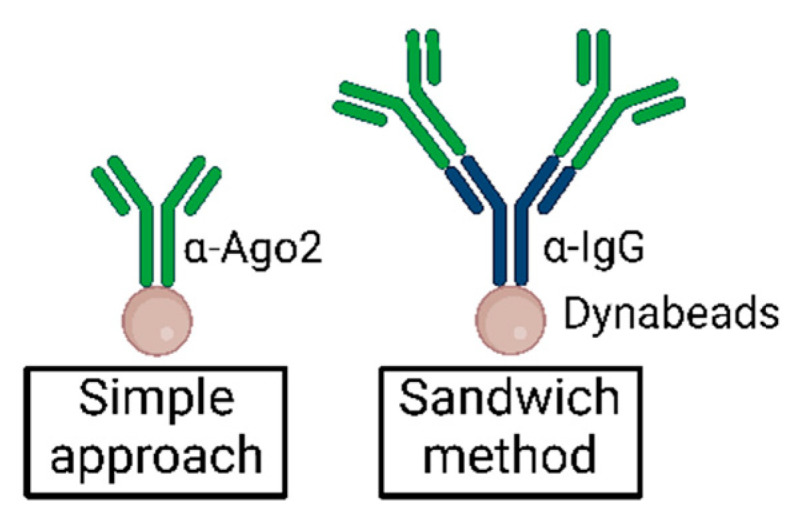
Simplified illustration of the sandwich method as compared to standard antibody–bead suspension in immunoprecipitation.

**Table 1 mps-05-00096-t001:** Suggested UV irradiation of cells.

Cell Line/Sample	4-Thiouridine (4-SU) Treatment	UV Wavelength (nm) ^1^	UV Irradiation (mJ/cm^2^)
Thin cell layer (e.g., A549)	−	254	150–300
	+	365	
Clustered, dense cell patches (e.g., Calu3)	−	254	300–450
	+	365	

^1^ Ensure the use of appropriate bulbs (clear glass bulbs are indicative of 254 nm (UV-C), and frosted glass bulbs of 365 nm (UV-A)).

**Table 2 mps-05-00096-t002:** Suggested RNase dilution series for primary experiments.

RNase Concentration	Mastermix	Final Dilution Factor in Sample
Highest	No predilution	10^−2^
High	5 µL RNase I (100 U/µL) in 45 µL RQ1 buffer	10^−3^
Medium	1 µL 10^−3^ dilution in 9 µL RQ1 buffer	10^−4^
Low	1 µL 10^−3^ dilution in 99 µL RQ1 buffer	10^−5^
Lowest	1 µL 10^−3^ dilution in 999 µL RQ1 buffer	10^−6^

**Table 6 mps-05-00096-t006:** PNK Mix for Radioactive Labeling.

Volume Per Reaction (µL)	Reagent
1.00	10× T4 PNK reaction buffer
x	[gamma-32P]-ATP + stable ATP
0.50	T4 PNK
0.25	RNaseOUT (40 U/µL)
y	RNase-free water
Total volume: 10.00	

**Table 9 mps-05-00096-t009:** Ligation Mix for Second Linker Ligation (3’ End of cDNA).

Volume Per Reaction (µL)	Reagent
0.30	Water
2.00	10× NEB RNA ligase buffer (with DTT)
0.20	100 mM ATP
9.00	50% PEG8000
0.50	High conc. RNA ligase (NEB M0437)
Total volume: 12.00	

**Table 15 mps-05-00096-t015:** Volumes for ProNex Size Selection.

Volume Per Reaction (µL)	Reagent
82.80	ProNex beads
300.00	Wash buffer
22.00	Elution buffer

## Data Availability

Not applicable.

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
