# Peer review of "Studying miRNA–mRNA Interactions: An Optimized CLIP-Protocol for Endogenous Ago2-Protein"

_mps, 2022, doi:10.3390/mps5060096_

Round 1
Reviewer 1 Report
In this manuscript, the authors have streamlined the optimized Ago2-specific CLIP protocol. The authors aim to focus on- reducing cellular materials, lysis conditions, and better radioactive labeling of miRNA. The experimental design remains more or less similar to all other CLIP protocols.
Comments for authors:
· Line no. 14 “Ago2-specifc” typo error
· Line no. 95 “We have found that the Ago2-iCLIP protocol, while relying on the quality of the Ago2 95 antibody of choice, is also heavily dependent on the cell lines used.”
Which adherent cell line authors used in their protocol, and what was the cell culture condition? Except for cell confluency, all other significant details are missing.
· How does this protocol a better choice than the other non-radioactive CLIP protocols? If it has been described then the message is not very clear.
· Line no. 273 and 275; Figure-1 legend is confusing in terms of box color.
· Some new references are missing that are published in late 2021 (Hafner M. et al., 2021 Nature Reviews) and 2022.
Author Response
In this manuscript, the authors have streamlined the optimized Ago2-specific CLIP protocol. The authors aim to focus on- reducing cellular materials, lysis conditions, and better radioactive labeling of miRNA. The experimental design remains more or less similar to all other CLIP protocols.
We agree with the reviewer that the difference is not large in comparison to many other published CLIP protocols. Our goal was to emphasize on the detailed changes that come with the application of CLIP protocols to the Ago2 protein. When we started our own Ago2-CLIP projects, we had problems to find a comprehensive, detailed manual for specifically Ago2-CLIP approaches, and we want to fill this gap with this manuscript. We aim to provide the opportunity for other groups to simplify the optimization for individual cell lines and to reduce their workload on the initial testing efforts and optimizations we ourselves had to go through significantly.
Comments for authors:
- Line no. 14 “Ago2-specifc” typo error
Thank you, this was corrected.
- Line no. 95 “We have found that the Ago2-iCLIP protocol, while relying on the quality of the Ago2 95 antibody of choice, is also heavily dependent on the cell lines used.”
Which adherent cell line authors used in their protocol, and what was the cell culture condition? Except for cell confluency, all other significant details are missing.
This was indeed missing from the manuscript, thank you for spotting this mistake. We have added the cell lines to line 95 and the cell culture conditions to paragraph 3.2.1 (line 323-325) in the “Procedure” section.
- How does this protocol a better choice than the other non-radioactive CLIP protocols? If it has been described then the message is not very clear.
The comparison between classical CLIP methodologies using ³²P-labeling of the RNA, and fluorescent linker ligation-based detection of crosslinked and immunoprecipitated RNAs is not focus of this methods manuscript. Our group has not yet used the fluorescent-linker ligation method, and we do not feel confident in describing or publishing this method. Nonetheless, since more and more institutes do not have access to isotope laboratories, the reviewers comment is of crucial importance. We have included a reference to the most recent non-radioactive CLIP protocol in the beginning, as well as in the very end of the manuscript (PMID: 33503264; Anastasakis et al, 2021, Nucleic Acids Res).
- Line no. 273 and 275; Figure-1 legend is confusing in terms of box color.
We would like to thank the reviewer for pointing this out. We have rephrased and re-structured the figure legend to make it clearer to the reader.
- Some new references are missing that are published in late 2021 (Hafner M. et al., 2021 Nature Reviews) and 2022.
Thank you, we have added the suggested reference in the appropriate parts of the text. We had initially overlooked this comprehensive review. This helps the reader to get more insight of that many different CLIP methodologies published to date. We have also included the most recent PAR-CLIP methods publication from 2022 that was missing (Danan et al, 2022, Methods Mol Biol), as well as mentioned the non-radioactive CLIP approach, as mentioned in a previous reviewer’s comment (Anastasakis et al, 2021, Nucleic Acids Res),
New references:
- Hafner, M., Katsantoni, M., Köster, T. et al. CLIP and complementary methods. Nat Rev Methods Primers 1, 20 (2021). https://doi.org/10.1038/s43586-021-00018-1
- Danan C, Manickavel S, Hafner M. PAR-CLIP: A Method for Transcriptome-Wide Identification of RNA Binding Protein Interaction Sites. Methods Mol Biol. 2022;2404:167-188. doi:10.1007/978-1-0716-1851-6_9
- Anastasakis DG, Jacob A, Konstantinidou P, et al. A non-radioactive, improved PAR-CLIP and small RNA cDNA library preparation protocol. Nucleic Acids Res. 2021;49(8):e45. doi:10.1093/nar/gkab011
---
Additionally, we have found and corrected two mistakes in Figures 4, 5 and 6, and have corrected the missing [kDa] labeling.
Reviewer 2 Report
The authors present a step-by-step procedure for precipitating miRNAs and mRNA-miRNA constructs using the RNA-binding protein Ago2. This protocol is well-written and thoroughly described. I simply have one little suggestion that could improve the work. At the conclusion, the possibility of using IgG antibodies for signal enrichment was mentioned. It would be most beneficial and beneficial to the work if this comparison was also experimentally proven.
This, though, is a minor point. Overall, it's a pretty successful paper.
Author Response
The authors present a step-by-step procedure for precipitating miRNAs and mRNA-miRNA constructs using the RNA-binding protein Ago2. This protocol is well-written and thoroughly described.
We would like to thank the reviewer for the positive evaluation of our manuscript, and for the constructive criticism/suggestion below.
I simply have one little suggestion that could improve the work. At the conclusion, the possibility of using IgG antibodies for signal enrichment was mentioned. It would be most beneficial and beneficial to the work if this comparison was also experimentally proven.
This, though, is a minor point. Overall, it's a pretty successful paper.
Thank you for this suggestion. We had considered adding an additional figure to the manuscript. We have decided against that in the end, since there is no difference in our hands, so in the given experimental setup, cell line and lysis method. But his is an important point, so we have added a paragraph discussing this (line 857-859). We have added the figure here for the reviewer, there is no difference in the efficiency of immunoprecipitation of Ago2-miRNA complexes:
[Figure: see attached PDF]
Figure for the reviewer. Initial testing of bridging antibody approach. Radiography of CLIP membrane, using Protein marker IV (pre-stained) peqGOLD. A549-ACE2 cells, crosslinked at 254 nm with 300 mJ/cm2 and lysed in 4× packed cell volume Lysis buffer. For IP, Dynabeads Protein G were incubated with 0.5 µg α-Ago2 clone 11A9, or 1.2 µg α-IgG (rat) followed by 0.5 µg α-Ago2 clone 11A9 (bridging antibody approach). RNase I digest employing dilutions of 10-3 and 10-5 was conducted followed by 5’ end dephosphorylation using rSAP before radioactive labeling with ATP mix (37.5% volume [gamma-32P]-ATP). Ago2-miRNA complexes run at 130 kDa.
---
Additionally, we have found and corrected two mistakes in Figures 5 and 6, and have corrected the missing [kDa] labeling.
